# Lessons from South Korea Regarding the Early Stage of the COVID-19 Outbreak

**DOI:** 10.3390/healthcare8030229

**Published:** 2020-07-24

**Authors:** Min Cheol Chang, Jong Hyun Baek, Donghwi Park

**Affiliations:** 1Department of Rehabilitation Medicine, Yeungnam University, Daegu 38541, Korea; wheel633@ynu.ac.kr; 2Department of Thoracic and Cardiovascular Surgery, Yeungnam University, Daegu 38541, Korea; yumc2000@yu.ac.kr; 3Department of Physical Medicine and Rehabilitation, Ulsan University Hospital, College of Medicine, University of Ulsan, Ulsan 44610, Korea

**Keywords:** coronavirus, COVID-19, drive-through, South Korea, China

## Abstract

South Korea has experienced difficulty in controlling the spread of the novel coronavirus disease (COVID-19) during the early stages of the outbreak. South Korea remains passionately determined to protect Koreans against COVID-19 and through trial and error hopes to improve the strategies used to limit the outbreak. Here, we review how COVID-19 spread and what prevention strategies were implemented during the early stages of the outbreak in South Korea. We investigated online newspapers published in South Korea from 21 January 2020 to 20 March 2020, and reviewed academic medical articles related to COVID-19. Additionally, we acquired data on COVID-19 cases through the official website for COVID-19 in South Korea. To date, numerous measures have been applied by the government and the medical community during the early stages of the COVID-19 outbreak including the reporting of methods for diagnostic testing, patient classification, the introduction of drive-through screening centers, COVID-19 preventive measures, implementation of government policies for the shortage of face masks, and entry restrictions. Here, we present data from the early stages of the COVID-19 outbreak and measures to prevent its spread in South Korea. We believe that sharing the experience of South Korea during the COVID-19 outbreak can help other countries to implement strategies to prevent its rapid transmission.

## 1. Introduction

Beginning in late December 2019, an outbreak of a novel coronavirus disease (COVID-19) occurred in Wuhan, China, and the disease subsequently spread to over 100 countries within two months [1]. On 11 March 2020, the World Health Organization (WHO) declared the outbreak a pandemic. Approximately 2% of COVID-19 cases result in death due to massive alveolar damage and progressive respiratory failure [2]. Although this fatality rate is low relative to that of the severe acute respiratory syndrome (SARS) (10%) and Middle East respiratory syndrome (MERS) (34%) coronaviruses, the fatality rate among elderly patients or those with underlying disorders is substantially greater [3]. Furthermore, because it is extremely contagious, many people are affected by COVID-19, and the infection rate continues to increase sharply [4]. Moreover, the number of deaths due to COVID-19 has exceeded that of SARS and MERS and will continue to increase [5]. The COVID-19 pandemic poses a threat to the general public health worldwide, causing a global economic and health crisis [6].

South Korea is geographically close to China, and many people migrate, travel, and commute between the two countries for pleasure or business. In February 2020, a COVID-19 outbreak occurred in South Korea immediately following the large outbreak in China. Human transmission of the severe acute respiratory syndrome coronavirus 2 (SARS-CoV-2), which causes COVID-19, occurs through infected respiratory droplets [2]. SARS-CoV-2 belongs to the subfamily Orthocoronavirinae, which characteristically binds with high affinity to mucous membranes and expresses spike proteins densely embedded on the phospholipid envelope [7]. Mutations in the genes that encode these spike proteins resulted in a 50-fold increase in the affinity for mucus compared with that of other coronaviruses [7,8]. This increases the chance of respiratory infections even when a small amount of the virus encounters mucus. In addition, the incubation period is approximately 5 days on average, during which infection occurs [9]. These characteristics of SARS-CoV-2 have resulted in an exponential increase in cases of COVID-19. Similar to many other countries, South Korea has never encountered such a situation. Therefore, South Korea has experienced difficulty in controlling the spread of COVID-19 during the early stages of the outbreak. South Korea remains passionately determined to protect its people against COVID-19, and through reviewing early efforts, we hope to improve prevention strategies.

Here, we review the early stages of the COVID- 19 outbreak, specifically investigating data on screening and diagnosis of cases and prevention methods, including governmental policies in South Korea. Furthermore, sharing South Korea’s experience with other countries confronting COVID-19 may help to shape strategies to prevent its rapid transmission.

## 2. Methods

We reviewed online newspapers (both national and local) published in South Korea from 21 January 2020 to 20 March 2020, and medical studies related with COVID-19 through Pubmed. We conducted a content analysis. We acquired data on COVID-19, such as the number of confirmed cases, death rate, and positive rate of COVID-19 tests from South Korea’s official website for COVID-19 (www.coronaboard.kr). There were no funds or time allocated for patient and public involvement; thus, we were unable to involve patients. We have invited patients to help us develop our dissemination strategy.

## 3. Results

### 3.1. Spread of COVID-19 in the Early Stages (2 Months After the First Case) of the Outbreak in South Korea

On 21 January 2020, the first case of COVID-19 was confirmed in South Korea in a patient with a history of travel to Wuhan. Subsequently, the first 28 confirmed cases of SARS-CoV-2 infection had a history of travel to China or contact with confirmed COVID-19 patients. On 16 February 2020, the 29th case was diagnosed with no history of overseas travel or contact with confirmed patients. One month after the appearance of the first case, 45 more COVID-19 cases were confirmed. By February 20, the number of new confirmed cases was greater than 50 per day and COVID-19 began to spread widely. In Daegu, South Korea, a rapid surge of COVID-19 occurred in late February and early March; over 5000 cases were confirmed as COVID-19. The main source of this rapid propagation was determined to be a group meeting of the religious group Shincheonji in Daegu on Sunday. In this meeting, thousands of people sat in close proximity to each other, in limited space within the religious facility, having conversations, praying, and singing songs. Nearly all the people (about 99%) in this religious group underwent a COVID-19 test, and 62% (over 2000 cases) tested positive for the virus. Between 27 February and 7 March, over 500 new cases per day of COVID-19 were reported in South Korea (Figure 1). 

Subsequently, until 13 March, the number of new confirmed cases of COVID-19 in South Korea was maintained in the range of 100–300 per day (Figure 1). After 14 March, the number of new confirmed cases decreased to less than 100 per day, except for one day (March 19: 152 cases). In addition, mass infections were frequently occurring in various facilities, such as churches, hospitals, nursing homes, and call centers. As of 20 March 2020, in South Korea, 8652 people (M:F = 3330:5322) had been confirmed with COVID-19. People in the age group of 20–29 years were greatly affected, accounting for 27.3% of the total COVID-19 cases, followed by those in the age groups of 50–59, 40–49, 60–69, 30–39, 70–79, 10–19, 80–89, and < 10 years (Figure 1). Of these cases, 100 patients have died (death rate: 1.16%) (Figure 2).

Out of these 100 patients, 92 (92%) were in the age groups 60–69, 70–79, and ≥ 80 (Figure 2) and the corresponding death rates in these age groups were 1.5%, 6.5%, and 11.6%, respectively. In contrast, although the largest number of cases infected by SARS-CoV-2 occurred in individuals in the 20–29 group, there were no fatalities among this population. As of 20 March 2020, a total of 6319 people were receiving treatment for COVID-19, and 2233 people have been released from isolation. In South Korea, as of 20 March 2020, 301,139 people have undergone COVID-19 testing, and 8652 people have tested positive. Therefore, the rate of positive test results was 2.9% (Figure 3).

### 3.2. Diagnostic Test for COVID-19 in South Korea

As of 20 March 2020, Koreans could undergo a diagnostic test for COVID-19 in 581 hospital or clinics, which were evenly distributed nationwide. There are three types of diagnostic tests for coronaviruses: a molecular diagnostic method (real-time polymerase chain reaction, RT-PCR), a culture method, and an antigen-antibody test method [10,11]. The culture method involves directly testing and culturing the virus for 2 to 7 days, which is both dangerous and ineffective for confirmation. The antigen-antibody test uses immunochromatography [12]. Diagnosis kits for this method provide results within 20 minutes, but the accuracy is less than 70%, which is not useful for confirmatory tests [11,13].

The final confirmatory test recommended by the WHO is RT-PCR, which identifies the presence of viral transcripts. Currently, PCR methods are the standard technique used by countries around the world, and WHO. The pancoronavirus RT-PCR assay has the highest sensitivity for diagnosing COVID-19, because it detects all known coronaviruses [13]. However, the process for pancoronavirus RT-PCR is complicated, and it takes more than 24 hours to be completed. In contrast, RT-PCR detects SARS-CoV-2 more efficiently by amplifying a specific gene that appears only in SARS-CoV-2, and provides results within six hours [11].

There are four diagnostic kits approved in South Korea, all of which use RT-PCR. Allplex™ SARS-CoV-2 Assay (Seegene®, South Korea) uses primers of the E gene (encapsulated protein gene), RNA-dependent RNA polymerase (RdRP) gene, and N gene (nucleocapsid protein gene). PowerChek^TM^ SARS-CoV-2 RT- PCR Kit (Kogene biotech®, South Korea) uses primers of 229E-specific genes, OC43-specific genes, and NL63-specific genes. DiaPlexQTM Novel Coronavirus Detection Kit (SolGent®, South Korea) uses primers of the Orf 1a and N genes [11]. Finally, STANDARD M n-CoV Real-Time Detection Kit (SD Biosensor®, South Korea) uses primers of the RdRp and E genes. In a study of 1014 patients in Wuhan who underwent both RT-PCR testing and chest computed tomography (CT) for evaluation of COVID-19, a "positive" chest CT for COVID-19 (as determined by consensus by two radiologists) had a sensitivity of 97%, using the PCR tests as a reference [14]. After approval by the Korean Food and Drug Administration, RT-PCR is applied in all coronavirus diagnostic tests, and the quality of the tests is monitored through the Korean Society for Diagnostic Medicine. In South Korea, according to the infectious disease prevention law, there is no examination fee for individuals with suspected SARS-CoV-2 infection, as determined by a physician. For those without suspected COVID-19 who still want to be tested, the fee is approximately USD 120 [11,15].

### 3.3. Classification System of Patients with COVID-19 in South Korea

In South Korea, during the early COVID-19 epidemic, all confirmed patients were hospitalized. However, due to a large rise in the number of patients with COVID-19, not all patients could be hospitalized. Several deaths occurred among patients self-isolated at home. To prevent these situations, on 2 March 2020, the Korea Centers for Disease Control and Prevention (KCDC) announced a four-stage patient classification system based on clinical symptoms, vital signs, chronic underlying disease, and age (Table 1). As of February 2020, the number of negative pressure inpatient treatment beds in South Korea was 1077.

Underlying diseases include diabetes, chronic kidney disease, chronic liver disease, chronic lung disease, chronic cardiovascular disease, blood cancer, cancer patients with chemotherapy, patients administered immunosuppressants, and human immunodeficiency virus disease.

Patients with chronic underlying diseases were defined as those with “diabetes, chronic kidney disease, chronic liver disease, chronic lung disease, chronic cardiovascular disease, blood cancer, cancer patients with chemotherapy, patients administered immunosuppressants, and human immunodeficiency virus disease.” Based on these criteria, cases were classified by disease severity. Patients without any symptoms or with mild disease were isolated in hotels or facilities designated by the government. Patients with moderate disease were admitted to infectious disease hospitals or nationally designated hospital isolation wards (negative pressure intensive care unit), and those with severe disease were admitted to nationally designated hospital isolation wards (negative pressure intensive care unit). The entire cost for isolation and treatment of COVID-19 was paid for by the government.

### 3.4. Drive-Through Screening Centers for COVID-19

As the number of symptomatic or suspected individuals increased, more efficient and safe screening systems became necessary. Thus, drive-through screening centers were designed and implemented in Korea, based on the previous concept of point of dispensing (POD) for bioterrorism and drive-through clinics for pandemic influenza [16]. In South Korea, the drive-through screening center for COVID-19 was first implemented on 23 February 2020 at the Kyungpook National University Chilgok Hospital, Daegu, where a large COVID-19 outbreak emerged [16]. The flow of the drive-through screening center is as follows: entrance, registration, examination, specimen collection, instructions, exit (Figure 4) [16].

The entire service was provided to symptomatic or suspected individuals without exiting their cars. The drive-through system, proven to be safe and efficient for COVID-19 screening, was implemented at 68 of the 577 COVID-19 screening centers in Korea (as of 12 March 2020) [16]. In other countries, such as Australia, Japan and the United States, drive-through screening centers were already implemented to cope with the global COVID-19 outbreak.

### 3.5. COVID-19 Preventive Measures

The following “COVID-19 Preventive Measures” were established by the government and shared with the public [17]: (1) frequent hand washing with soap and water for at least 20 seconds; (2) avoid touching of eyes, nose, and mouth; (3) avoid close contacts with people who are sick; (4) avoid visiting crowded places; (5) use of face masks when visiting hospitals or clinics; (6) cough or sneeze into a tissue; (7) conferences, seminars, or meetings not allowed at work.

### 3.6. Measures for the Shortage of Face Masks

Due to a severe shortage of face masks, on 26 February 2020, the government banned export of face masks from South Korea, and took full control of face mask distribution via public channels (pharmacies). The government implemented a 5-day rotation system for mask distribution [17]. Mask purchases were limited to two per week on designated days of the week based on the last digit of the year of birth on ID cards (e.g., driver’s license) (Monday: 1 or 6, Tuesday: 2 or 7, Wednesday: 3 or 8, Thursday: 4 or 9, Friday: 5 or 0). Purchase was available during the weekends for those who could not buy masks on weekdays. The price was fixed to USD 1.30 per mask. Buying masks for others and multiple purchases in a single day were not permitted. For medical doctors, the government supplied 2–5 masks a week per person. Masks were locally produced in South Korea.

### 3.7. Country Entry Restrictions

Even two months after the first COVID-19 case, South Korea had not adopted country entry restrictions from foreign countries, despite the recommendations from other groups, such as the Korean National Medical Association. In contrast, other countries have implemented policies prohibiting entry or quarantining individuals for two weeks upon entry from Korea as well as China. In addition, unlike Wuhan, China, the government did not enforce isolation in any areas of Korea. The Korean government only enforced cohort quarantine in cases of infectious outbreaks in patients in certain medical facilities or nursing homes, such as the Cheongdo dae-nam hospital. Despite self-containment guidelines for Koreans by the Chinese local government (which appears to be the source of COVID-19) as well as opposition from various groups in South Korea, South Korea continues to maintain the above policy.

## 4. Discussion

The population of South Korea was 51,225,308 in 2019, and the land area covers about 100,000 square kilometers; therefore, South Korea is a densely populated country. Additionally, approximately 90% of the total population lives in an urban environment. Considering that SARS-CoV-2 is mainly spread through person-to-person droplets, South Korea is particularly vulnerable to SARS-CoV-2 transmission [18]. Moreover, South Korea is geographically close to China. Due to these factors, South Korea faced a serious outbreak of COVID-19. In the early stages of the COVID-19 outbreak in South Korea due to mass infections (e.g., Shincheonji), confirmed cases of COVID-19 surged rapidly. Additionally, mass infections from churches, hospitals, nursing homes, and call centers were largely responsible for the increased number of COVID-19 cases. Therefore, during the outbreak of a virus with high transmissibility, group meetings should be restricted and measures to actively prevent COVID-19 should be applied in crowded facilities, such as hospitals and nursing homes.

Individuals in their twenties accounted for the greatest number of SARS-CoV-2 infections. However, most patients in this age group had mild symptoms and were occasionally asymptomatic. Interestingly, no patients in the 20–29 group died from COVID-19 [2]. We think that young people may be careless regarding infection or spread of SARS-CoV-2 virus, because they are mostly resistant to infection. Because asymptomatic patients are more likely to transmit the disease, and fatality rates are higher for elderly individuals, young individuals may significantly contribute to preventing COVID-19 transmission by considering even mild symptoms, apply social distancing rules, and through other preventative measures.

The refinement of the COVID-19 diagnostic capacity in South Korea was due to the approval of the rapid COVID-19 test kits using RT-PCR. These kits have the advantage of quickly diagnosing cases within six hours compared to the pancoronavirus RT-PCR, making it possible to test more people. However, most kits use only 2–3 primers for SARS-CoV-2 gene detection. Considering that coronaviruses are prone to mutation, using kits that rely on only a few primers may lead to false negative results. Therefore, although a negative result cannot explicitly rule out COVID-19, the kit’s diagnostic capacity has helped greatly to manage COVID-19 in South Korea.

In the early stage of the COVID-19 outbreak, South Korea faced a lack of hospital beds (negative pressure intensive care units) for COVID-19 patients. Therefore, to prevent a similar situation, early planning to supply and allocate hospital beds is essential. In most countries, there are insufficient hospital beds (negative pressure intensive care unit) for COVID-19 patients. Therefore, similar to South Korea, it is necessary to classify patients according to disease severity and allocate beds or hospitalize, mainly, patients with severe symptoms. Furthermore, when a shortage of face masks is anticipated, it is recommended that the government take control of the supplies for their constant and even distribution. 

Many countries have applied entry restrictions, despite the potential negative impact on global trade and diplomatic relations. Evidence on the effect of entry restrictions for blocking the influx of COVID-19 is lacking. Several previous studies reported on its limited effectiveness for controlling highly contagious viral outbreaks [19,20]. In 2014, Mateus et al. conducted a meta-analysis examining 23 relevant studies to evaluate the effect of entry restrictions implemented around the world during the 2009 H1N1 influenza pandemic [19]. Entry restrictions reduced the incidence of influenza cases by less than 3%. They concluded that the entry restrictions were not helpful in controlling influenza outbreak in any particular area. However, they reported that entry restrictions delayed progression of infectious diseases for weeks or months. South Korea did not implement entry restrictions in the early stage of the COVID-19 outbreak, which could be associated with the rapid spread of COVID-19 that was observed. In our opinion, entry restrictions may provide time to prepare hospital beds and medical equipment for COVID-19 patients. Therefore, it is recommended that each country must evaluate the potential benefits of implementing entry restrictions after accurate analysis of their national economy, diplomatic relations, and COVID-19 preparedness. 

In the early stages of the COVID-19 outbreak in South Korea, more females (*n* = 5322) were infected by SARS-CoV-2 than males (*n* = 3330). Considering the fact that numbers of confirmed cases in other countries were not significantly different between males and females, the imbalanced sex ratio in Shincheonji (the religious group in which mass infections occurred) would likely account for the higher number of confirmed cases in females.

## 5. Conclusions

In conclusion, here we examined the COVID-19 prevalence in South Korea and broadly characterized the population affected during the early transmission of COVID-19, as well as the measures implemented to prevent COVID-19 spread during the early stages of the outbreak. We believe that the experience of South Korea can help other countries to shape strategies to prevent rapid transmission of COVID-19. However, our study is not a systematic review; in the future, strategic and systematic reviews including data during the COVID-19 pandemic and measures or efforts of the government and the medical community are warranted. 

## Figures and Tables

**Figure 1 healthcare-08-00229-f001:**
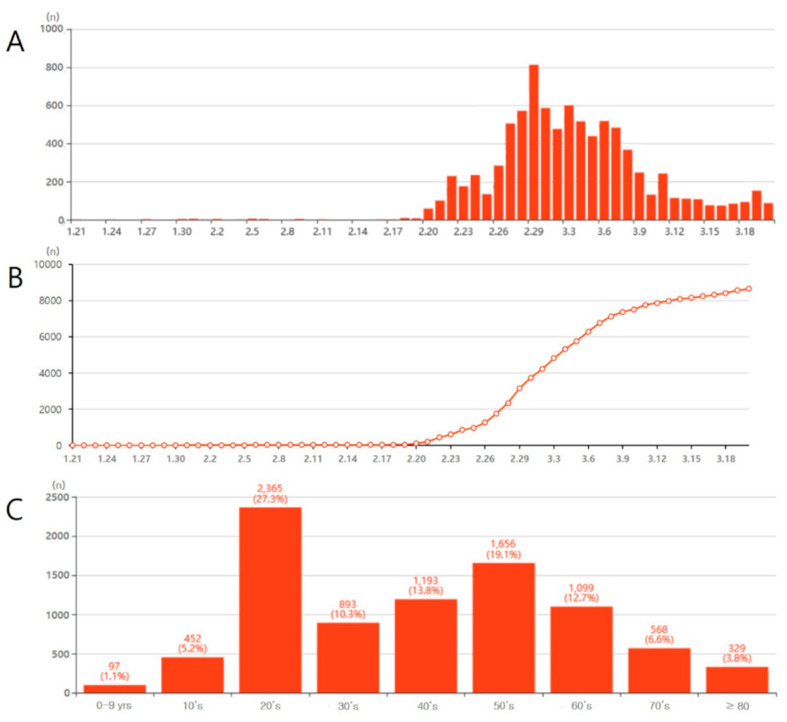
(**A**) Daily number of coronavirus disease (COVID-19) confirmed cases in South Korea, (**B**) daily cumulative number of COVID-19 confirmed cases in South Korea, (**C**) number of COVID-19 confirmed cases according to age range. Source: www.coronaboard.kr.

**Figure 2 healthcare-08-00229-f002:**
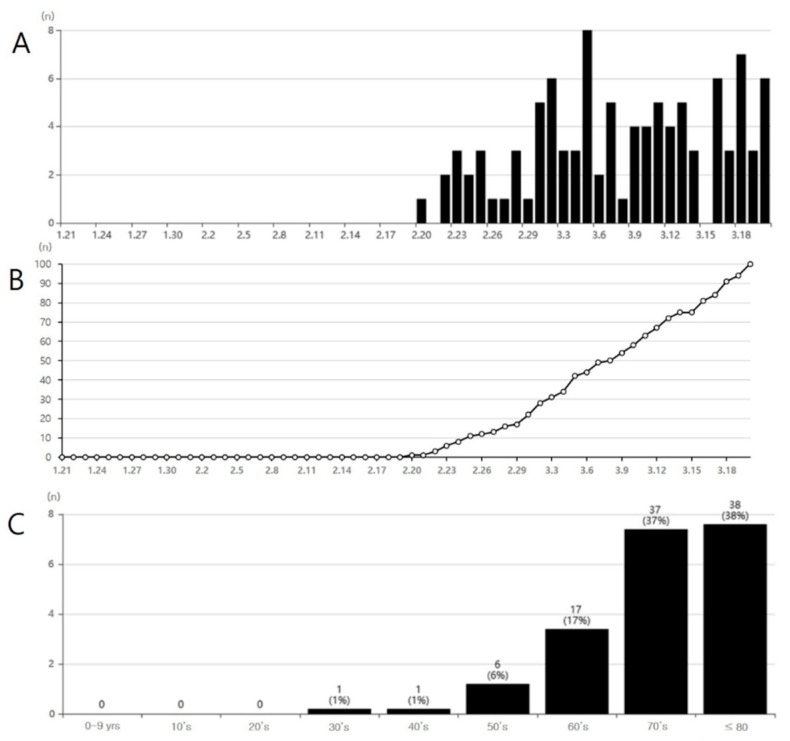
(**A**) Daily number of deaths due to COVID-19 in South Korea, (**B**) daily cumulative number of deaths due to COVID-19 in South Korea, (**C**) number of deaths due to COVID-19 according to age. Source: www.coronaboard.kr.

**Figure 3 healthcare-08-00229-f003:**
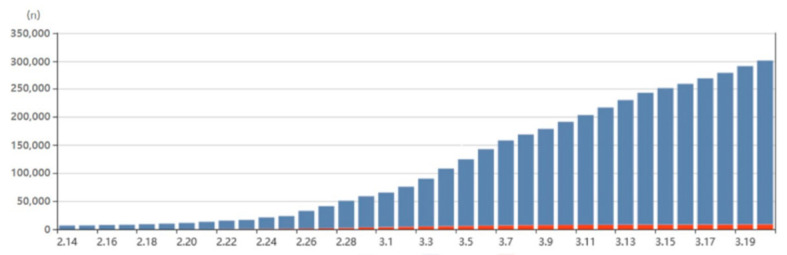
Cumulative numbers of positive and negative test results for severe acute respiratory syndrome coronavirus 2 (SARS-CoV-2). In total, 301,139 people underwent testing for SARS-CoV-2, and 8,652 (2.9%) tested positive. (red: cases confirmed as SARS-CoV-2 positive; blue: cases confirmed as SARS-CoV-2 negative) Source: www.coronaboard.kr.

**Figure 4 healthcare-08-00229-f004:**
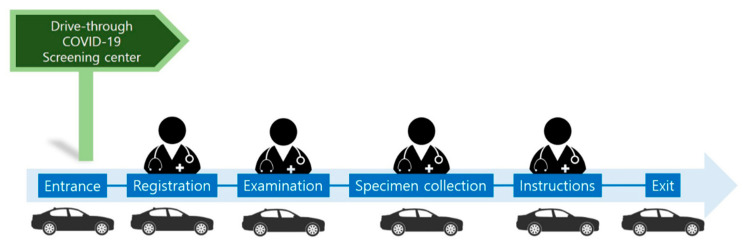
Schematic illustration of a drive-through screening center for COVID-19 testing.

**Table 1 healthcare-08-00229-t001:** Classification of patients with COVID-19 and protection methods.

Classification	Classification Criteria	Protection Methods
Asymptomatic	All of the following conditions are met. (1) Clear consciousness, (2) < 50 years old, (3) no underlying disease, (4) non-smoker, and (5) fever < 37.5 °C without antipyretics	Isolation in hotels, which were adopted and allocated by the government
Mild	Clear consciousness and one or more of the following conditions are met (1) > 50 years old, (2) ≥ 1 underlying disease, and (3) fever ≤ 38 °C with antipyretics	
Moderate	Clear consciousness and one or more of the following conditions are met (1) fever >38 °C even with antipyretics, (2) respiratory difficulty	Infectious disease hospital, nationally designated hospital isolation ward (negative pressure intensive care unit)
Severe	Unconscious state	Nationally designated hospital isolation ward (negative pressure intensive care unit)

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
