# Peer review of "Lessons from South Korea Regarding the Early Stage of the COVID-19 Outbreak"

_healthcare, 2020, doi:10.3390/healthcare8030229_

Round 1

Reviewer 1 Report

The submitted article “Lessons from the Situation of South Korea regarding the early stage of COVID-19 outbreak” by Min Cheol Chang et al. provides unique insights into the handling of the pandemic and the difficulties of early outbreak management within a specific setting. In geographical vicinity to China and densely populated across its high degree of urbanization, South Korea is particularly vulnerable to COVID-19 transmission. The paper reflects the challenges and management strategies in the early stage of the epidemic, i.e. in the two months after the first case occurred in South Korea.

The text aims at communicating these challenges and their handling to the scientific community, including targeted recommendations. No border entry restriction, the appearance of mass infections and a notable high level of infections of people in their 20s, among other specificities,  make South Korea indeed an interesting case for comparison with other countries.

The provided insights and recommendations offer relevant hints for management strategies facing outbreaks of corona viruses or other epidemics in the future, as well as for the evaluation of previous outbreak responses.

Methodologically the authors draw on newspaper reports, COVID-19 studies and official COVID-19 data provided by the government – all within the paper’s focus time frame of the early outbreak period (January 21 to March 20).

All authors occupy positions in the medical field though with no self-evident connections to the topic of the paper. It might be interesting to learn more about their work encounters with, and interest in, the corona pandemic.

The unique country case as presented here, focused in time and space, is the actual strength of this paper. This might be further enhanced by a) a more detailed description in some parts, b) by reflecting some cross-connections within the presented material, and c) by contrasting the presented country case with international figures. Here are some corresponding suggestions:

  • Table 1 lists “isolation in hotels” but not “self-isolation at home” as described in line 146.·        
  • Please explain the “isolation in hotels”. Were hotels adapted and allocated by the government especially for this purpose?
  • The exact number of beds in “negative pressure intensive care units” in Table 1 would allow for a convenient comparison with other countries.
  • The gender distribution of confirmed cases (line 92) is striking (M:F 3,330:5,322). How do you explain this high proportion of women? Is this uneven sex ratio also seen in mass infections (line 78, 91-92), in the age distribution of confirmed cases and deaths, in the classification according to the severity of cases and in the death cases (line 95)?
  • If masks were only mandatory when visiting hospitals or clinics (line 185), how do you explain the shortage of masks (line 188-196)? Furthermore, please distinguish and be more precise describing masks for the general population and masks for the medical personal. Please also specify if masks were locally produced or if they had to be imported (from where?).·
  • Please remark with the first usage of the term 2019-nCoV (line 44) that this older term for the virus was replaced by the now internationally accepted designation SARS-CoV-2.

The English of the paper is appropriate and understandable. A careful reading, however, would help to clean up some of the very few minor mistakes and disadvantageous formulations. E.g. line 186 “Conferences, seminars, or meeting[s is missing] are not to be held at work”.

Author Response

Reviewer 1

We really appreciate your kind comments for our manuscript.

  • Table 1 lists “isolation in hotels” but not “self-isolation at home” as described in line 146.·        

Answer: After elucidating the strategies and measures, patients who confirmed as COVID-19 was not self-isolated at home. They were isolated in hotel or hospital. However, prior to the elucidation of formal strategy, some patients were isolated at home. Therefore, it seems that any correction is not necessary.

  • Please explain the “isolation in hotels”. Were hotels adapted and allocated by the government especially for this purpose?

Answer: It was adapted and allocated by the government. We added this content to the manuscript.

  • The exact number of beds in “negative pressure intensive care units” in Table 1 would allow for a convenient comparison with other countries.

Answer: We presented the exact number of negative pressure treatment beds.

  • The gender distribution of confirmed cases (line 92) is striking (M:F 3,330:5,322). How do you explain this high proportion of women? Is this uneven sex ratio also seen in mass infections (line 78, 91-92), in the age distribution of confirmed cases and deaths, in the classification according to the severity of cases and in the death cases (line 95)?

Answer: I think that the different sexual ratio in our nation was due to mass infection of Sincheonji religious group. We added this matter to the discussion. Also, it is not definitely demonstrated fact, we suggested further evaluation on this matter. Regarding the age distribution of deaths and the classification according to the severity of cases, we could not obtain data related to those matters.

  • If masks were only mandatory when visiting hospitals or clinics (line 185), how do you explain the shortage of masks (line 188-196)? Furthermore, please distinguish and be more precise describing masks for the general population and masks for the medical personal. Please also specify if masks were locally produced or if they had to be imported (from where?).·

Answer: The people in Korea could only buy only 2 masks a week per 1 person due to shortage of masks. The patients could enter hospitals or clinics with wearing those masks. Medical doctors received masks from the government. Masks were locally produced, not imported. We added the contents related with this matter to the manuscript.

  • Please remark with the first usage of the term 2019-nCoV (line 44) that this older term for the virus was replaced by the now internationally accepted designation SARS-CoV-2.

The English of the paper is appropriate and understandable. A careful reading, however, would help to clean up some of the very few minor mistakes and disadvantageous formulations. E.g. line 186 “Conferences, seminars, or meeting[s is missing] are not to be held at work”.

Answer: We corrected 2019-nCoV into SARS-CoV-2. Our manuscript was edited by the native English speaker again following the reviewer’s comments.

Reviewer 2 Report

HEALTHCARE - 869505 -peer-review-v1

Dear authors,

While this is a very interesting paper that will help countries dealing with COVID-19, though, as it is written, it cannot be published.

I hope the following suggestions will help you to improve your work.

GENERAL COMMENTS:

  1. Please follow the authors instructions. Example references in brackets and not in parenthesis.

  1. Needs proof reading

  1. At some points the writing is “journalistic” and not academic. Example in the abstract (the first thing someone reads): “In conclusion, we reviewed the data from…” (the method we used was..?)

  1. There are sentences and points that need references. Example: line 38-39 & 39-40 (and many more).

  1. The full stop after the brackets.

INTRODUCTION

  1. Line 42: migrate only? or do they also travel and commute due to their jobs and businesses?

  1. Line 44-45: “The human transmission of the 2019 novel coronavirus (2019-nCoV), which causes COVID-19, is through infected respiratory droplets.(2)” IS what? Something is missing (spread?)

  1. At the end of the introduction provide with the scope and objectives (if any) of the paper.

  1. line 57 : we review.. the question is why review? What is the aim of this paper? To describe what has been implemented in South Korea with specific objectives the measures and restrictions?

  1. Then address how you are going to tackle the issue

  1. Why is this paper important (it is a very interesting as a paper, but you need to state the importance too, especially for the countries experiencing the same problem)

METHODS

  1. Methods: needs more in-depth analysis
  2. You reviewed newspapers. How many?
  3. National ones (on national level) or and smaller ones that are on domestic community level?

  1. Newspapers from what date to what date.
  2. How was the analysis done? Content analysis?
  3. Also you wrote about newspapers that you were reviewing (see line 62)

  1. Line 63: by reviewing previous studies: example???

  1. If you did both then write it so it will be clear. We used data from three / two/ four different bases. 1…..2…3…. . specifically, for …. We …..

  1. Line 66 what is PPI???

RESULTS

  1. Are these from content analysis of newspapers or from literature review? Where did these come from?

  1. Figures: source. If it is yours then: Source: The authors (based on content analysis of..?)

  1. Line 103: currently: Today we have 15-7-2020. specify date and probably update

  1. 5 & 3.6 : references

  1. See throughout the paper where references are needed. You refer to things that are known to the medical community of Korea but are not known to the rest of the world, and thus references are needed. (e.g. 124-126, 134-142)

DISCUSSION:

  1. In the discussion section we also compare to what other academics have found- or other countries have done.

There are many examples of academic papers from Asia, but from other countries too:

Example of papers are:

Wu, Z., & McGoogan, J. M. (2020). Characteristics of and important lessons from the coronavirus disease 2019 (COVID-19) outbreak in China: summary of a report of 72 314 cases from the Chinese Center for Disease Control and Prevention. Jama, 323(13), 1239-1242.

Kamenidou, I. E., Stavrianea, A., & Liava, C. (2020). Achieving a Covid-19 Free Country: Citizens Preventive Measures and Communication Pathways. International Journal of Environmental Research and Public Health, 17(13), 4633.

 Fosu, G. O., & Edunyah, G. (2020). Flattening The Exponential Growth Curve of COVID-19 in Ghana and Other Developing Countries; Divine Intervention Is A Necessity. Divine Intervention Is A Necessity (March 31, 2020).

Leiblfinger, M., Prieler, V., Schwiter, K., Steiner, J., Benazha, A., & Lutz, H. Resources to support community and institutional Long-Term Care responses to COVID-19.

Knafo, W. (2020). Monitoring the propagation of COVID-19-pandemic first waves. medRxiv.

Atre, S. R. (2020). Lessons from COVID-19 in India: Extended lockdowns–At what cost?. Medical Journal of Dr. DY Patil Vidyapeeth, 13(3), 192.

I hope that these comments will help you with your work

Best of luck with your paper and be safe!

Author Response

Reviewer 2

Dear authors,

While this is a very interesting paper that will help countries dealing with COVID-19, though, as it is written, it cannot be published.

I hope the following suggestions will help you to improve your work.

GENERAL COMMENTS:

  1. Please follow the authors instructions. Example references in brackets and not in parenthesis.

Answer: We presented references with brackets.

  1. Needs proof reading

Answer: Our manuscript was edited by the native English speaker again following the reviewer’s comments.

  1. At some points the writing is “journalistic” and not academic. Example in the abstract (the first thing someone reads): “In conclusion, we reviewed the data from…” (the method we used was..?)

Answer: Following the reviewer’s comment, we corrected the contents in Abstract.

  1. There are sentences and points that need references. Example: line 38-39 & 39-40 (and many more).

Answer: We added more references to each sentence in the manuscript.

  1. The full stop after the brackets.

Answer: We corrected it.

INTRODUCTION

  1. Line 42: migrate only? or do they also travel and commute due to their jobs and businesses?

Answer: We added “travel and commute due to their jobs and businesses” to the manuscript.

  1. Line 44-45: “The human transmission of the 2019 novel coronavirus (2019-nCoV), which causes COVID-19, is through infected respiratory droplets.(2)” IS what? Something is missing (spread?)

Answer: We inserted “spread” in between “is” and “through”.

  1. At the end of the introduction provide with the scope and objectives (if any) of the paper.

Answer: We described the scope and objectives in more detail.

  1. line 57 : we review.. the question is why review? What is the aim of this paper? To describe what has been implemented in South Korea with specific objectives the measures and restrictions?

Answer: We presented the aim of this paper in more detail. Specific objectives and measures were presented in results section. To write it to introduction can be redundant.

  1. Then address how you are going to tackle the issue

Answer: Following the reviewer’s comment, we revised the introduction section.

  1. Why is this paper important (it is a very interesting as a paper, but you need to state the importance too, especially for the countries experiencing the same problem)

Answer: We explained the importance of our paper to the end of the introduction section.

METHODS

  1. Methods: needs more in-depth analysis

Answer: Following the reviewer’s comments, we tried to describe method section in more detail.

  1. You reviewed newspapers. How many?

Answer: We tried to read nearly whole published newspaper, but unfortunately we did not count the number of them.

  1. National ones (on national level) or and smaller ones that are on domestic community level?

Answer: Both were included, and we presented this matter to the manuscript.

  1. Newspapers from what date to what date.

Answer: We reviewed online newspapers (both national and local) published in South Korea from January 21, 2020 to March 20, 2020. We already presented it.

  1. How was the analysis done? Content analysis?

Answer: I presented that we conducted content analysis.

  1. Also you wrote about newspapers that you were reviewing (see line 62)

Answer: We tried to read nearly whole published newspaper. Some of them are Korean. I am sorry but I think it is not necessary to the manuscript.

  1. Line 63: by reviewing previous studies: example???

Answer: By using Pubmed database, we reviewed researches. We presented this matter to the contents.

  1. If you did both then write it so it will be clear. We used data from three / two/ four different bases. 1…..2…3…. . specifically, for …. We …..

Answer: We described the data search in more detail.

  1. Line 66 what is PPI???

Answer: It is patient and public involvement. We presented it.

RESULTS

  1. Are these from content analysis of newspapers or from literature review? Where did these come from?

Answer: We added the source of the results of analysis. Also, if the results came from the previous studies, we added the references.

  1. Figures: source. If it is yours then: Source: The authors (based on content analysis of..?)

Answer: We presented the source of the analysis. 

  1. Line 103: currently: Today we have 15-7-2020. specify date and probably update

Answer: We correct the contents related to the reviewer’s comment.

  1. 5 & 3.6 : references

Answer: We added the references

  1. See throughout the paper where references are needed. You refer to things that are known to the medical community of Korea but are not known to the rest of the world, and thus references are needed. (e.g. 124-126, 134-142)

Answer: We added the references to the manuscript.

DISCUSSION:

  1. In the discussion section we also compare to what other academics have found- or other countries have done.

Answer: In this study, we described the specific measures the Korea government applied to this manuscript. However, the below articles is majorly on the number of infection or epidemiology. Therefore, I think that the topics of our manuscript and previous studies can be different. The main purpose of this study is introducing the measures and strategies of Korea government. In the future, it would be interesting topic to compare measures of each country.

There are many examples of academic papers from Asia, but from other countries too:

Example of papers are:

Wu, Z., & McGoogan, J. M. (2020). Characteristics of and important lessons from the coronavirus disease 2019 (COVID-19) outbreak in China: summary of a report of 72 314 cases from the Chinese Center for Disease Control and Prevention. Jama, 323(13), 1239-1242.

Kamenidou, I. E., Stavrianea, A., & Liava, C. (2020). Achieving a Covid-19 Free Country: Citizens Preventive Measures and Communication Pathways. International Journal of Environmental Research and Public Health, 17(13), 4633.

 Fosu, G. O., & Edunyah, G. (2020). Flattening The Exponential Growth Curve of COVID-19 in Ghana and Other Developing Countries; Divine Intervention Is A Necessity. Divine Intervention Is A Necessity (March 31, 2020).

Leiblfinger, M., Prieler, V., Schwiter, K., Steiner, J., Benazha, A., & Lutz, H. Resources to support community and institutional Long-Term Care responses to COVID-19.

Knafo, W. (2020). Monitoring the propagation of COVID-19-pandemic first waves. medRxiv.

Atre, S. R. (2020). Lessons from COVID-19 in India: Extended lockdowns–At what cost?. Medical Journal of Dr. DY Patil Vidyapeeth, 13(3), 192.

I hope that these comments will help you with your work

Best of luck with your paper and be safe!
